# The relaxation of residual inclusion pressure and implications to Raman-thermobarometry

Xin Zhong[*1]; Evangelos Moulas[2]; Lucie Tajčmanová[3]

1 Physics of Geological Processes, The Njord Center, University of Oslo, Oslo, Norway

5   2 Institut des sciences de la Terre, Université de Lausanne, Lausanne, Switzerland

Institute of Earth Sciences, Heidelberg University, Germany

*Correspondence to*: Xin Zhong (xinzhong0708@gmail.com)

**Abstract:** Residual pressure can be preserved in mineral inclusions, e.g. quartz-in-garnet, after exhumation due to differential expansion between inclusion and host crystals. Raman spectroscopy has been applied to infer the residual

pressure and provides information on the entrapment temperature and pressure conditions. However, the amount of residual pressure relaxation cannot be directly measured. An underestimation of pressure relaxation may lead to significant errors between calculated and actual entrapment pressure. This study focuses on three mechanisms responsible for the residual-pressure relaxation: 1) viscous creep; 2) plastic yield; 3) proximity of inclusion to thin-section surface. Criteria are provided to quantify how much of the expected residual pressure is relaxed due to these three mechanisms. An analytical solution is

introduced to demonstrate the effect of inclusion depth on the residual pressure field when the inclusion is close to thin-section surface. It is shown that for quartz-in-garnet system, the distance between thin-section surface and inclusion centre needs to be at least two times the inclusion radius to avoid pressure relaxation. In terms of viscous creep, representative case studies on quartz-in-garnet system show that viscous relaxation may occur from temperatures as low as 600~700 °C depending on the particular *P-T* path and various garnet compositions. For quartz entrapped along the prograde *P-T* path and

subject to viscous resetting at peak *T* above 600~700 °C, its residual pressure after exhumation may be higher than predicted from its true entrapment conditions. Moreover, such a viscous resetting effect may introduce apparent overstepping of garnet nucleation that is not related to reaction affinity.

## 1. Introduction

During metamorphism, the growth of porphyroblasts often results in the entrapment of inclusions, e.g. quartz inclusion

entrapped in garnet host. Due to the differences in the elastic compressibility and thermal expansion coefficient between the

inclusion and host, residual inclusion pressures may be preserved after exhumation (e.g. Rosenfeld and Chase, 1961; Gillet

et al., 1984; Zhang, 1998; Angel et al., 2015). The residual pressure can be inferred by Raman shift based on experimental

calibrations, e.g. quartz inclusions (e.g. Liu and Mernagh, 1992; Schmidt and Ziemann, 2000). This allows the application of

Raman-thermobarometry to infer the entrapment $P$-$T$ conditions (e.g. Ashley et al., 2014; Bayet et al., 2018; Enami et al.,

2007; Izraeli et al., 1999; Kohn, 2014; Spear et al., 2014; Taguchi et al., 2019; Zhong et al., 2019). Existing models that link

residual pressure and entrapment $P$-$T$ conditions are based on elastic rheology and often assume infinite host radius (e.g.

Rosenfeld and Chase, 1961; Van Der Molen and Van Roermund, 1986; Guiraud and Powell, 2006; Angel et al., 2017).

Recent experimental works have been successfully performed to compare the measured residual pressure with modelled

residual pressure under well-controlled $P$-$T$ conditions for synthetic samples with quartz-in garnet system (Thomas and

Spear, 2018).

Although many studies using Raman spectroscopy reported residual pressure close to the predictions from elastic model (e.g.

Ashley et al., 2014; Enami et al., 2007; Zhong et al., 2019), a large amount of inclusion pressure estimates are lower than

theoretically predicted by the elastic model (Korsakov et al., 2009; Kouketsu et al., 2016; Yamamoto et al., 2002). The

relaxation of inclusion pressure can be due to various reasons and a systematic investigation is critical to better apply

Raman-thermobarometry to natural samples. Meanwhile, Raman-thermobarometry has been employed to investigate the

amount of overstepping for garnet growth by comparing the $P$-$T$ constraints from phase equilibria and elastic

thermobarometry (Spear et al., 2014; Wolfe and Spear, 2017). The relaxation of residual inclusion pressure may lead to

errors in the calculated reaction affinities (e.g. Castro and Spear, 2017).

When a mineral inclusion maintains residual pressure, differential stress is concomitantly developed around the inclusion on

the host side to maintain mechanical equilibrium (e.g. Zhang, 1998; Tajčmanová et al., 2014). The host mineral may

experience viscous creep which is manifested by the dislocation structures (Chen et al., 1996; Yamamoto et al., 2002).

Furthermore, the host mineral may also form radial/tangential (micro)-cracks due to plastic yield when the differential stress

exceeds the yield criterion (Van Der Molen and Van Roermund, 1986; Whitney, 1991). Mechanical models show that both viscous creep (dislocation or diffusion creep of host) and plastic yield (radial or tangential micro-cracks) can cause

significant pressure relaxation (Dabrowski et al., 2015; Zhang, 1998). This would lead to an underestimate of residual inclusion pressure (Zhong et al., 2018b) (Fig. 1). Meanwhile, during the thin-section preparation, mineral inclusions are positioned into proximity towards the thin-section surface (Fig. 1). The thin-section surface is stress free and may elastically relax the residual pressure (Mindlin and Cheng, 1950; Seo and Mura, 1979; Zhong et al., 2018a). It is of petrological interest to study how deep the inclusion needs to be in order to preserve the residual pressure. Experimental works and numerical

simulations with finite element method have been performed to test the safe inclusion depth (inclusion radius less than one half of host radius) so that the residual inclusion pressure can be preserved for the application of Raman barometry (Campomenosi et al., 2018; Mazzucchelli et al., 2018).

In this contribution, we systematically investigate the following mechanisms for residual pressure relaxation: 1) creep of the host materials that causes viscous relaxation of residual pressure, 2) plastic yield that causes (micro)-cracks that relax the

residual pressure and 3) relaxation due to the proximity of inclusion towards thin-section surface. For the first and second purposes, a 1D visco-elasto-plastic mechanical model is developed in radially symmetric spherical coordinate frame to study the effect of viscous relaxation and plastic yield of the residual entrapment pressure. The derived system of equations is nondimensionalized to extract the key parameters that control the amount of viscous relaxation and plastic yield of the residual pressure. For the third purpose, an analytical solution for the entrapment pressure field close to thin-section surface

is introduced as a simplified form based on the work of Seo and Mura (1979). The analytical solution demonstrates the effect of the inclusion depth that controls the amount of pressure relaxation. This solution applies to the case where the inclusion possesses the same elastic moduli as the host, and the stress is generated due to the differential thermal expansion/contraction. In comparison, for natural quartz-in-garnet system, numerical solutions are applied to investigate the safe distance that causes negligible pressure relaxation due to the presence of thin-section surface. In this study, both

inclusion and host are treated as elastically isotropic as an assumption to put focus on the effect of these three mechanisms on preserved residual pressure. The effects of elastic anisotropy for commonly encountered quartz inclusion have been





studied experimentally and theoretically by e.g. Murri et al. (2018) and Campomenosi et al. (2018) and are discussed in the

end.

## 2. Methods

**2.1 Visco-elasto-plastic mechanical model**

To investigate the effect of viscous creep and plastic yield on residual pressure, we develop a 1D mechanical model with

spherical symmetry that is based on the conservation of mass and momentum, and it employs a Maxwell visco-elasto-plastic

rheology. In 1D radially symmetric spherical coordinate frame, mechanical equilibrium is expressed as follows:

$$\frac{\partial \tau_{rr}}{\partial r} + \frac{3\tau_{rr}}{r} - \frac{\partial P}{\partial r} = 0, \tag{1}$$

where $\tau_{rr}$ is the radial component of deviatoric stress ($Pa$), $P$ is pressure ($Pa$) and $r$ is radial coordinate ($m$). We apply the

Maxwell visco-elasto-plastic rheology to express stress-strain (rate) relation in the radial direction as follows:

$$\dot{e}_{rr} = \underbrace{\frac{\dot{\tau}_{rr}}{2G}}_{elastic} + \underbrace{\frac{\tau_{rr}}{2\eta}}_{viscous} + \underbrace{\lambda\ sign(\tau_{rr})}_{plastic}, \tag{2}$$

where the dot above $\dot{\tau}_{rr}$ denotes time derivative, $\dot{e}_{rr}$ is the radial component of the deviatoric strain rate ($s^{-1}$) that is

composed of elastic, viscous and plastic counterparts, $G$ is shear modulus ($Pa$), $\eta$ is viscosity ($Pa \cdot s$), $\lambda$ is the plastic

multiplier ($s^{-1}$) which guarantees that the plastic yield criterion is not exceeded. The plastic strain rate is obtained by using

the Tresca yield criterion (see e.g. Ranalli, 1995):

$$F = |\tau_{rr} - \tau_{tt}| - C, \tag{3}$$

where $C$ is plastic cohesion ($Pa$) that controls the occurrences of (micro)-cracks, and $\tau_{tt}$ is the tangential component of

deviatoric stress. Due to spherical symmetry, we also have $\tau_{tt} = -1/2\tau_{rr}$. Applying the plastic flow law (e.g. Vermeer and

De Borst, 1984), we get:





$$\dot{e}_{rr}^{p} = \lambda \frac{\partial F}{\partial \tau_{rr}} = \lambda \, sign(\tau_{rr}), \begin{cases} \lambda = 0 \text{ for } F \leq 0 \\ \lambda \neq 0 \text{ for } F > 0 \end{cases}. \tag{4}$$

The non-Newtonian (effective) viscosity is expressed as follows:

$$\eta = A|\tau_{rr}|^{1-n}, \tag{5}$$

where $A$ is the temperature-dependent pre-factor for viscosity ($Pa^n \cdot s$), $n$ is the stress (power-law) exponent. The pressure

can be expressed as a function of volume and temperature via equation of state (EoS), and its time derivative is as follows:

$$\dot{P} = -\dot{\varepsilon}_{kk}/\beta + \alpha\dot{T}/\beta, \tag{6}$$

where $\beta$ is compressibility ($1/Pa$), $\alpha$ is the thermal expansion coefficient ($1/K$), $\dot{T}$ is the rate of temperature change ($K/s$). Temperature is treated as homogeneous within inclusion-host system. Einstein summation is used here for the volumetric strain rate ($\dot{\varepsilon}_{kk} = \dot{\varepsilon}_{rr} + 2\dot{\varepsilon}_{tt}$). No viscous or plastic volumetric strain is considered. This assumption is a good approximation for non-porous, crystalline materials (e.g. Moulas et al., 2019).

By applying first-order finite difference in time to Eq. 2 and Eq. 6 (i.e. $\dot{\tau}_{rr} = \frac{\tau_{rr} - \tau_{rr}^{o}}{\Delta t}$ and $\dot{P} = \frac{P - P^{o}}{\Delta t}$), we can explicitly express $\tau_{rr}$ and $P$ as:

$$\tau_{rr} = 2\eta Z\dot{e}_{rr} + (1 - Z)\tau_{rr}^{o} - 2\eta Z\lambda \, sign(\tau_{rr}), \tag{7}$$

$$P = P^{o} - \Delta t\dot{\varepsilon}_{kk}/\beta + \alpha\Delta t\dot{T}/\beta, \tag{8}$$

where $Z = \frac{\Delta t G}{\Delta t G + \eta}$ is the viscoelastic coefficient, $\tau_{rr}^{o}$ is the radial component of deviatoric stress in the previous time step, $P^{o}$ is the pressure in previous time step. If the yield criterion in Eq. 3 is exceeded ($F > 0$), the plastic multiplier must be correctly chosen to drive $F$ to zero. This can be achieved by substituting the deviatoric stress (Eq. 7) into Eq. 3 and let $F = 0$.

Therefore, we obtain $\lambda$ as follows:

$$\lambda = \delta\dot{e}_{rr} + \frac{(1-Z)sign(\tau_{rr})}{2\eta Z}\tau_{rr}^{o} - \frac{C}{3\eta Z}, \quad \text{if } F > 0 \text{ (otherwise } \lambda = 0). \tag{9}$$





**2.2 Nondimensionalization**

The variables in the above equations can be scaled to derive nondimensional parameters for better understanding the behaviour of the inclusion-host system. This is done by choosing the following independent scales: the inclusion radius $R$, temperature change $\Delta T$, time $t^*$, viscosity pre-factor $A_h$ of host, plastic cohesion $C_h$ of host, and the expected pressure

perturbation $P_{exp}$ that is given as follows:

$$P_{exp} = \frac{\Delta P(\beta_i - \beta_h) - \Delta T(\alpha_i - \alpha_h)}{\beta_i + 3/4G_h}, \tag{10}$$

where $\Delta P, \Delta T$ are the confining pressure and temperature drops from entrapment to the Earth's surface, $\beta_i$ and $\beta_h$ are the compressibility of inclusion and host, $\alpha_i$ and $\alpha_h$ are the thermal expansion coefficients of inclusion and host, $G_h$ is the shear modulus of host.

By choosing $P_{exp}$ as the scale, residual pressure will vary around zero to one. This pressure scale allows convenient

quantification for viscous and plastic relaxation.

The involved physical variables are scaled as shown below:

$$r = R\,\bar{r}$$

$$\beta = \frac{1}{P_{exp}}\bar{\beta}$$

$$G = P_{exp}\bar{G}$$

$$\alpha = \frac{1}{\Delta T}\bar{\alpha}$$

$$P = P_{exp}\bar{P}$$

$$\dot{T} = \frac{\Delta T}{t^*}\bar{\dot{T}}$$

$$\tau_{rr} = P_{exp}\overline{\tau_{rr}}$$

$$C = C_h\bar{C}$$

(11)





$$\eta = P_{exp}t^*\bar{\eta}$$

$$F = P_{exp}\bar{F}$$

$$\Delta t = t^*\overline{\Delta t}$$

$$A = A_h\bar{A}$$

$$\lambda = \frac{1}{t^*}\bar{\lambda}$$

$$v_r = \frac{R}{t^*}\overline{v_r}$$

where the overhead bars indicate dimensionless properties. Substituting these scaling equations into Eq. 1, 7 and 8, we get:

$$\frac{\partial\overline{\tau_{rr}}}{\partial\bar{r}} + \frac{3\overline{\tau_{rr}}}{\bar{r}} - \frac{\partial\bar{P}}{\partial\bar{r}} = 0, \tag{12}$$

$$\bar{P} = \overline{P^o} + \frac{1}{\bar{\beta}}\left[-\overline{\Delta t}\frac{\partial\bar{r}^2\overline{v_r}}{\bar{r}^2\partial\bar{r}} + \bar{\alpha}\bar{T}\right], \tag{13}$$

$$\overline{\tau_{rr}} = \frac{4}{3}\bar{\eta}\bar{Z}\left(\frac{\partial\overline{v_r}}{\partial\bar{r}} - \frac{\overline{v_r}}{\bar{r}}\right) + (1-\bar{Z})\overline{\tau_{rr}^o} - 2\bar{\eta}\bar{\lambda}\delta\bar{Z}, \tag{14}$$

where dimensionless viscosity, viscoelastic coefficient and plastic multiplier are expressed as:

$$\bar{\eta} = De \cdot \bar{A}|\overline{\tau_{rr}}|^{1-n}, \tag{15}$$

$$\bar{Z} = \frac{\overline{\Delta t}\bar{G}}{\overline{\Delta t}\bar{G} + \bar{\eta}}, \tag{16}$$

$$\bar{\lambda} = \frac{4}{3}\delta\left(\frac{\partial\overline{v_r}}{\partial\bar{r}} - \frac{\overline{v_r}}{\bar{r}}\right) + \frac{(1-\bar{Z})\delta}{2\bar{\eta}\bar{Z}}\overline{\tau_{rr}^o} - C^* \cdot \frac{\bar{C}}{3\bar{\eta}\bar{Z}}, \text{ if } \frac{3}{2}\delta\overline{\tau_{rr}} - C^* \cdot \bar{C} > 0. \tag{17}$$

Two dominant dimensionless numbers emerge after nondimensionalization. They are Deborah number $De$ and Cohesion

number $C^*$ defined as follows:

$$De = \frac{A_h/P_{exp}^n}{t^*}, \tag{18}$$





$$C^* = \frac{C_h}{P_{exp}}.$$ (19)

where $A_h$ is the pre-factor of viscosity of the host, $n$ is stress exponent, $t^*$ is the duration of viscous relaxation, $C_h$ is the cohesion of host.

The Deborah number ($De$) is the ratio between the characteristic viscous relaxation time ($A_h/P_{exp}^n$) and model duration ($t^*$). If $De > 1$, the system behaves in an elastic manner, and if $De < 1$, viscous creep becomes important. The pre-factor of

viscosity is temperature dependent. By choosing the pre-factor $A_h$ at peak temperature, one can directly use $De$ to estimate the maximal amount of viscous relaxation. This is especially suitable for the process of isothermal decompression in many high-pressure rocks.

The Cohesion number $C^*$ characterizes the ability of a host mineral to plastically yield and a high Cohesion number implies that the material is less prone to plastic yield. The viscosity of different mineral phases may vary by orders of magnitude,

and the cohesion of difference mineral may also vary by many several factors, potentially orders of magnitude. Therefore, these two dimensionless numbers have a dominant effect on the amount of inclusion pressure modification due to viscous relaxation and plastic yield.

**2.3 Numerical approach for visco-elasto-plastic model**

The dimensionless viscosity (Eq. 15), viscoelastic coefficient (Eq. 16) and plastic multiplier (Eq. 17) can be substituted into

pressure equation (Eq. 13) and deviatoric stress equation (Eq. 14). Together with mechanical equilibrium equation (Eq. 12), they form a system of three equations with three unknowns, namely $\bar{v}_r$, $\overline{\tau_{rr}}$ and $\bar{P}$. Because viscosity, viscoelastic coefficient and plastic multiplier are functions of deviatoric stress, the system of equations is nonlinear. We solve for the three variables using an iterative method. Within the iteration loop, an elastic test stress is first evaluated by letting $\bar{\lambda} = 0$ so that the prediction for the yield function $\bar{F}$ is computed. If $\bar{F} < 0$, no plastic yield occurs and $\bar{\lambda}$ is remains zero. Otherwise the

prediction of the yield function is positive and $\bar{\lambda}$ is computed based on Eq.17 to drive $\bar{F}$ back to zero. The elastic moduli are updated based on pressure and temperature conditions from tabulated look-up tables within the iteration. The look-up tables are pre-computed based on EoS. We used the EoS for quartz crystal from Angel et al. (2017a), and the EoS for pyrope,



grossular and almandine crystals from Milani et al. (2015). The detailed expressions of EoS can be found in the EoSFit7c

software documentation (Angel et al., 2014). The EoS for spessartine is from Gréaux and Yamada, (2014). The

compressibility and thermal expansion coefficient for garnet are averaged based on the molar percentage of garnet

endmembers. The shear moduli of garnet endmembers are from Bass (1995). The host radius is set to be 10 times the

inclusion radius to make boundary effects negligible. Temperature is treated as homogeneous in space. After the iteration

loop, the residuals of the three equations 12, 13 and 14 are minimized below ca. $10^{-12}$. The numerical model has been

benchmarked using the analytical solution with elastic, non-Newtonian viscous rheology in Zhong et al., 2018. The

numerical benchmark for elasto-plastic rheology is performed by using the analytical solution of Zhang, (1998) (see

supplementary materials).

### 2.4 Analytical solution of inclusion pressure close to thin-section surface

Pressure relaxation takes place when the inclusion is brought into proximity to a stress-free thin-section surface. Mindlin and

Cheng (1950) provided a closed-form analytical solution of stress field inside and outside a spherical inclusion with thermal

strain in a semi-infinite host. The analytical solution has been generalized to ellipsoidal inclusion (Seo and Mura, 1979).

Substantial mathematical investigations have also been done in deriving the analytical solution of the elastic field for

inclusion in half-space (e.g. Tsuchida and Nakahara, 1970; Aderogba, 1976; Jasiuk et al., 1991). In this work, a simplified

analytical formulation of pressure field within a spherical inclusion $P_{inc}$ close to thin-section surface is given. It is

emphasized that in this situation the inclusion and host possess the same elastic moduli and the residual pressure is caused

only by thermal expansion/contraction. The goal here is to analytical demonstrate the effect of inclusion's proximity to the

thin-section surface. Cartesian coordinate system is employed as shown in Fig. 2. The full stress tensor $\sigma_{ij}$ of inclusion

loaded with eigenstrains is represented as follows (Seo and Mura, 1979).

$$\sigma_{ij} = \frac{\varepsilon^*(1+\nu)G}{2\pi(1-\nu)}\left[-4\pi\delta_{ij} - \frac{\partial^2\psi}{\partial x_i x_j} + 4\nu\delta_{ij}\frac{\partial^2\phi}{\partial x_3 x_3} + (3-4\nu)(\delta_{3j}+\delta_{3j}-1)\frac{\partial^2\phi}{\partial x_i x_j} - \right.$$

$$\left.(\delta_{3j}+\delta_{3j})\frac{\partial^2\phi}{\partial x_i x_j} - 2x_3\frac{\partial^3\phi}{\partial x_3 x_i x_j}\right]. \tag{20}$$

While for the host, stresses are given below





$$\sigma_{ij} = \frac{\varepsilon^*(1+\nu)G}{2\pi(1-\nu)}\left[-\frac{\partial^2\psi}{\partial x_i x_j} + 4\nu\delta_{ij}\frac{\partial^2\phi}{\partial x_3 x_3} + (3-4\nu)(\delta_{3j}+\delta_{3j}-1)\frac{\partial^2\phi}{\partial x_i x_j} - (\delta_{3j}+\right.$$
$$\left.\delta_{3j})\frac{\partial^2\phi}{\partial x_i x_j} - 2x_3\frac{\partial^3\phi}{\partial x_3 x_i x_j}\right], \tag{21}$$

where the indices of $x_i$ ($i = 1,2,3$) are in Cartesian coordinate frame following the order of $x$, $y$ and $z$ (see Fig. 2), and $\varepsilon^*$ is

the isotropic eigenstrain that is expressed as the difference of volumetric strain between inclusion and host assuming that

they are not bounded by each other. As the inclusion and host possess the same elastic moduli, the difference of volumetric

strain is only caused by the thermal expansion coefficient difference.

$$\varepsilon^* = -\frac{\Delta T(\alpha_i - \alpha_h)}{3}. \tag{22}$$

The elliptic integrals $\psi$ and $\phi$ are expressed below:

$$\psi = \pi R^3 \int_\lambda^\infty \frac{1 - \frac{R_1^2}{R^2+s}}{(R^2+s)^{\frac{3}{2}}} ds, \tag{23}$$

where $\lambda = R_1^2 - R^2$ for host, $\lambda = 0$ for inclusion, and $R_1 = \sqrt{x_1^2 + x_2^2 + (x_3 - L)^2}$.

$$\phi = \pi R^3 \int_\lambda^\infty \frac{1 - \frac{R_2^2}{R^2+s}}{(R^2+s)^{\frac{3}{2}}} ds, \tag{24}$$

where $\lambda = R_2^2 - R^2$ for both host and inclusion, and $R_2 = \sqrt{x_1^2 + x_2^2 + (x_3 + L)^2}$. Here, we focus on inclusion and derive a

simplified form for the pressure of inclusion. For the inclusion, the elliptic integrals are derived:

$$\psi = 2\pi(R^2 - \frac{1}{3}R_1^2), \tag{25}$$

$$\phi = \frac{4}{3}\pi R^3 R_2^{-1}. \tag{26}$$

The normal stresses in the inclusion are:





$$\sigma_{11} = \frac{\varepsilon^*(1+\nu)G}{2\pi(1-\nu)}\left[-4\pi - \frac{\partial^2\psi}{\partial x_1 x_1} + 4\nu\frac{\partial^2\phi}{\partial x_3 x_3} - (3-4\nu)\frac{\partial^2\phi}{\partial x_1 x_1} - 2x_3\frac{\partial^3\phi}{\partial x_3 x_1 x_1}\right], \tag{27}$$

$$\sigma_{22} = \frac{\varepsilon^*(1+\nu)G}{2\pi(1-\nu)}\left[-4\pi - \frac{\partial^2\psi}{\partial x_2 x_2} + 4\nu\frac{\partial^2\phi}{\partial x_3 x_3} - (3-4\nu)\frac{\partial^2\phi}{\partial x_2 x_2} - 2x_3\frac{\partial^3\phi}{\partial x_3 x_2 x_2}\right], \tag{28}$$

$$\sigma_{33} = \frac{\varepsilon^*(1+\nu)G}{2\pi(1-\nu)}\left[-4\pi - \frac{\partial^2\psi}{\partial x_3 x_3} + 4\nu\frac{\partial^2\phi}{\partial x_3 x_3} + (3-4\nu)\frac{\partial^2\phi}{\partial x_3 x_3} - 2x_3\frac{\partial^3\phi}{\partial x_3^3} - 2\frac{\partial^2\phi}{\partial x_3^2}\right]. \tag{29}$$

By substituting $\psi$ and $\phi$ into the equations above, the normal stresses can be obtained. In deriving the pressure, i.e. negative mean stress, many terms in Eq. 27-29 can be cancelled out. A simplified form is obtained as follow:

$$P_{inc} = \frac{4\varepsilon^*(1+\nu)G}{3(1-\nu)}\left[1 - \frac{2}{3}\frac{R^3}{R_2^3}(1+\nu)\left(\frac{3(z+L)^2}{R_2^2}-1\right)\right]. \tag{30}$$

Substituting the eigenstrain $\varepsilon^*$ and the expression of $P_{exp}$ in Eq. 10 into pressure, we obtain:

$$P_{inc} = P_{exp}\left[1 - \frac{2}{3}\frac{R^3}{R_2^3}(1+\nu)\left(\frac{3(z+L)^2}{R_2^2}-1\right)\right]. \tag{31}$$

The equation can be nondimensionalized by using $R$ as length scale shown below:

$$\frac{P_{inc}}{P_{exp}} = 1 - \frac{2}{3}\frac{1+\nu}{\bar{R}_2^3}\left(\frac{3(\bar{z}+\bar{L})^2}{\bar{R}_2^2}-1\right). \tag{32}$$

The analytical solution for pressure in the mineral inclusion subject to an initial residual pressure $P_{exp}$ is obtained. When the inclusion is far from thin-section surface ($\bar{L} \to +\infty$, and $\bar{R}_2 \to +\infty$), the actual residual pressure approaches the expected residual pressure based on classical elastic model ($P_{inc} \to P_{exp}$). The pressure field of an inclusion in half space based on Eq.

32 is shown in Fig. 2 using the Poisson ratio $\nu$ of pyrope crystal.

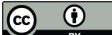

## 3. Results

### 3.1 Systematic investigation on Deborah number and Cohesion number

The amount of inclusion pressure relaxation is systematically investigated for the two inelastic deformation mechanisms (i.e. viscous creep and plastic yield) as a function of $De$ and $C^*$. At the beginning of the model, a residual pressure with the

inclusion is prescribed, and the far-field host maintains zero confining pressure. The pre-factor of viscosity is fixed as temperature does not vary. The purpose of this diagram in Fig. 3 is to systematically demonstrate how much the initially prescribed residual pressure can be relaxed due to viscous creep and plastic yield as controlled by $De$ and $C^*$. This diagram may assist petrological investigations because $De$ and $C^*$ can be evaluated based on experimental rock deformation data for different minerals, and they can be used with the diagram to check if viscous relaxation and plastic yield are expected or not.

The computed residual inclusion pressure is shown in Fig. 3. The thickness of plastic yield region is plotted as contours. The thick grey contour represents the onset of plastic yield starting from the inclusion-host interface and propagating towards the host side (Fig. 3). Based on the amount of inclusion pressure relaxation, three regimes are distinguished.

*Elastic regime* takes place when $De$ and $C^*$ are higher than one. Under these circumstances, no viscous relaxation and plastic yield occurs. The residual inclusion pressure is close to the expected residual pressure ($P_{inc} \approx P_{exp}$). This regime is the most

suitable for the application of Raman-thermobarometry.

*Viscous regime* dominates when $De$ is lower than one, and $C^*$ is above the plastic onset shown by the thick grey contour. In this case, the main mechanism responsible for the inclusion pressure relaxation is viscous creep. The effect of stress exponent on the amount of viscous relaxation is also significant. In general, a higher stress exponent delays pressure relaxation (c.f. Dabrowski et al., 2015). As the viscosity of natural minerals is low at high temperature conditions, viscous

regime may be reached at high temperature that leads to the relaxation of residual pressure.

*Plastic regime* prevails when $C^*$ is lower than one, and $De$ is located above the plastic onset. Under this circumstance, the residual pressure is not significantly relaxed by viscous creep, but by plastic yield. In general, the radius of plastic yield region is positively related to amount of residual pressure relaxation.

## 3.2 Pressure relaxation due to free surface

As a case study, the stress fields of quartz-in-almandine and almandine-in-quartz systems are numerically modelled using a finite difference (FD) thermo-elastic model (governing equations and model benchmark are provided in supplementary material). These examples are chosen to investigate two end-members: elastically stiffer host (quartz-in-almandine in Fig. 4a) and softer host (almandine-in-quartz in Fig. 4b). Pressures at three points within the inclusion (top, centre and bottom) are contoured as a function of $\bar{L}$ (see Fig. 4). The pressures evaluated at these three localities based on the analytical solution

in Eq. 32 are also shown by the dashed curves for comparison with numerical solutions. With decreasing distance to thin-section surface, the heterogeneity of pressure field increases. The pressure at the top point relaxes the most. Meanwhile, non-negligible pressure relaxation also takes place at the centre and bottom points. It is shown that pressure relaxation is less significant in elastically stiffer host (garnet) than in elastically softer host (quartz).

It is shown that the difference between analytical and numerical solution due to the difference of elastic moduli becomes

significant when the inclusion depth is shallow. The analytical solution and numerical solution are similar when evaluated at the bottom point at any depth. For quartz-in-garnet system, the analytical solution overestimates the amount of pressure relaxation (Fig. 4a). Assuming 3% pressure relaxation as acceptable for the application of Raman barometry, the analytical solution yields safe distance ca. $\bar{L}=2.0$ for the bottom and centre point, while the numerical solution yields ca. $\bar{L}=1.5$. For the top point, the safe distance ca. $\bar{L}=2.5$ based on the analytical solution is again higher than the prediction of ca. $\bar{L}=2.0$ based

on numerical solution. The difference of safe distance between analytical and numerical solution is due to the presence of elastically stiffer garnet host.

Differential stress is also shown in Fig. 4b. High differential stress at the host appears when the inclusion is close to thin-section surface. Differential stress may also exist inside the inclusion but it is in general smaller than that of the host. For quartz-in- garnet system, the differential stress forms a "ring" shaped pattern with a peak at the surface. The differential

stress may reach up to three times the expected residual pressure. This may potentially trigger plastic failure at thin-section surface. However, for the garnet-in-quartz system, such pattern is not observed even if the inclusion depth is shallow.

### 3.3 Viscous relaxation of quartz-in-garnet system

Assuming that the thin-section surface is sufficiently far away from a quartz inclusion and no microcracks appear around quartz inclusion, only viscous creep may contribute to the relaxation of residual pressure. Here, we show the effect of

viscous relaxation, particularly influenced by the temperature, on the preserved residual pressure. Using $De$ as a criterion to estimate the amount of viscous relaxation (Fig. 3), we show the relationship between temperature, inclusion pressure, and relaxation time given $De$=1 (see Eq. 18) in Fig. 5. The flow law of garnet from Wang and Ji (1999) is applied. The flow law parameters are given in the figure caption. The melting point of pyrope-rich garnet, grossular and spessartine are from Karato et al. (1995). For almost pure almandine, the garnet melting point is found to be 1588 K from Mohawk Garnet Inc,

which is slightly higher than 1570 K for almandine rich ($Alm_{0.68}Prp_{0.20}Grs_{0.12}$) garnet in Karato et al. (1995).

As an example, for a quartz inclusion possessing 0.5 GPa residual pressure maintained at 650 $^{\circ}$C, viscous relaxation will occur during 1 Ma for almandine rich garnet host. This temperature becomes higher (700 $^{\circ}$C) for pyrope rich garnet. If the residual pressure is used to recover the entrapment pressure given temperature higher than 650~700 $^{\circ}$C, an underestimate of the entrapment pressure may potentially occur.

In Fig. 6, synthetic retrograde $P$-$T$ paths from eclogite and amphibolite-facies metamorphic conditions are prescribed with different peak temperature. The entrapment $P$-$T$ conditions for the three synthetic $P$-$T$ paths are along elastic isomekes, which are the isopleths of residual inclusion pressure as a function of entrapment $P$-$T$ conditions. Therefore, the residual inclusion pressure should be the same if viscous relaxation is not considered. By involving viscous rheology of garnet host, different residual inclusion pressures are predicted. For the $P$-$T$ path starting at 800 $^{\circ}$C, 2 GPa, the quartz inclusion pressure

is predicted to be less than 0.2 GPa. The residual inclusion pressure subject to viscous relaxation is used to determine the apparent entrapment pressure (Fig. 6b). In Fig. 6b, it is shown that for the entrapment pressure within eclogite-facies conditions at 700 $^{\circ}$C and by using only the elastic model, a value of entrapment pressure is inferred that is approximately 10% less than the actual value. The amount of underestimate of entrapment pressure increases to 30% when the entrapment temperature reaches 800 $^{\circ}$C. The total exhumation time is set as 1 Ma.

For amphibolite-facies entrapment conditions, the residual pressure that is preserved in the quartz inclusion is significantly

       lower compared to the case where the entrapment occurred at eclogites-facies conditions. In this case, the amount of

       underestimate is less as well due to the fact that the viscosity of garnet host is stress dependent (see Eq. 5). As shown in Fig.

       6D, ca. 5% and 20 % underestimate of true entrapment pressure is predicted depending whether the entrapment occurred at

       700 $^{o}$C or 800 $^{o}$C, respectively.

**3.4 Pressure relaxation along prograde *P-T* path and apparent overstepping**

       The pressure relaxation problem is complicated when the quartz inclusion is entrapped not at the peak *P-T* conditions, but

       along the prograde *P-T* path. In this case, viscous relaxation occurs also along the prograde *P-T* path before the rock reaches

       the peak *P-T* conditions. Two synthetic *P-T* paths are illustrated in Fig. 7. In Fig. 7a, the quartz is entrapped in the

       almandine-garnet host at 400 $^{o}$C, 1 GPa and further experiences eclogites-facies *P-T* conditions. During the prograde path,

the quartz inclusion will develop underpressure, which will also be subject to viscous relaxation over geological time. The

       quartz pressure starts to converge towards the garnet host pressure at *T*>600 $^{o}$C. Nearly complete viscous resetting is

       observed when the system is brought up to 800 $^{o}$C. The prograde time is set as 1 Ma or 10 Ma to compare the amount of

       viscous relaxation as a function of time in Fig. 7.

       The rock may also stay at the peak *P-T* conditions before decompression occurs. A synthetic clockwise *P-T* path reaching

eclogite facies metamorphic condition is constructed as shown in Fig. 8. The quartz inclusion is entrapped into the garnet

       host at 400 $^{o}$C, 0.6 GPa, which is considered to be along the entrance of garnet stability field. Subsequently the system is

       brought to 700~750 $^{o}$C, 1.8~1.9 GPa conditions and stays there for 5 Ma. Afterwards, the retrograde *P-T* path takes 10 Ma.

       Two different *P-T* paths of quartz inclusions are constructed based on the implemented elastic and visco-elastic rheologies.

       Interestingly, the residual pressure of the inclusion that was subject to viscous relaxation is significantly higher (by 0.2 GPa,)

than the prediction of pure elastic model as shown by the black dashed curve (0.14 GPa). The apparent entrapment pressure

       is calculated using the predicted residual pressure for the inclusion whose host experienced viscous relaxation. A large

       discrepancy exists between the apparent entrapment pressure (ca. 1 GPa at the entrapment *T* 400 $^{o}$C) and the true entrapment

       pressure (0.6 GPa). The overall overestimate of true entrapment pressure (0.6 GPa) is about 0.3~0.4 GPa, which may

       potentially be interpreted as overstepping of the garnet growth/nucleation.





**4. Discussion**

**4.1 What may cause the residual pressure relaxation?**

The three mechanisms investigated here, i.e. viscous creep, plastic yield and proximity of inclusion to thin-section surface

can all be responsible for the relaxation of the residual inclusion pressure. The amount of inclusion-pressure relaxation due

to these three mechanisms is controlled by Deborah number ($De$), Cohesion number ($C^*$) and dimensionless depth ($\bar{L}$),

respectively. These three numbers are recommended to be examined beforehand.

In the examples of quartz-in-garnet systems, the residual pressure is considered to be sealed in creak-free garnet host.

However, cracks have been observed around some quartz inclusions but those inclusions are often avoided (e.g. Ashley et

al., 2014; Kouketsu et al., 2016). Based on our study, the presence (radius) of plastic yield region and preserved residual

inclusion pressure are dominated by Cohesion number ($C^* = C_h/P_{exp}$) as shown in Fig. 3. Cohesion $C_h$ can be converted

from hardness test data using the formula below (e.g. Evans and Goetze, 1979):

$$C_h = H/C_g \tag{33}$$

where $H$ is the measured microhardness and $C_g$ is a constant accounting for the indenter's geometry in the experiment.

Taking $C_g = 3$ (Evans and Goetze, 1979), the cohesion of garnet host is between 4.4 and 5 GPa at room conditions (Whitney

et al., 2007), which leads to a Cohesion number $C^* = 4.4{\sim}5$ given residual inclusion pressure $P_{exp} = 1$ GPa. This suggests

that plastic yield does not occur in an idealized scenario of isotropic, spherical quartz inclusion entrapped in infinite garnet

host. However, such an ideal scenario is highly improbable in natural samples. The observed cracks in garnet host may be

formed due to potential reasons including: 1) elevated differential stress when the inclusion is close to thin-section surface

("ring" shaped pattern in Fig. 4a); 2) stress concentration at the corners of quartz inclusion (Whitney et al., 2000); 3)

anisotropic elastic deformation of the quartz inclusion (e.g. Murri et al., 2018); 4) pre-fractures/weakness in garnet host

before the entrapment of quartz inclusions. Although our model does not predict exact conditions for plastic yield due to the

above possibilities, it gives a lower bound for the cohesion and provides information on what type of host mineral phase

cannot be used for Raman-barometry. Cohesion data of some common rock-forming minerals measured in hardness tests are



compiled and provided in table 1. As an example, given $P_{exp} = 1$ GPa, the Cohesion number of calcite host is ca. 0.6, and

dolomite is ca. 1.5 (Wong and Bradt, 1992). This implies that calcite will partially relax the residual pressure $P_{exp}$ and

dolomite has the potential to preserve $P_{exp}$. Care must be taken to check the potential presence of microcracks around

inclusions when using host minerals with low Cohesion number for Raman-barometry, e.g. zircon, dolomite. Minerals such

as calcite should be avoided to be used as the host material for the application of Raman barometry.

After thin-section preparation, the inclusion pressure may be (partially) relaxed. The dimensionless depth can be evaluated

by performing depth-step scan analysis with Raman spectroscopy in order to observe if the pressure gradually decreases

towards thin-section surface (Enami et al., 2007; Campomenosi et al., 2018). For quartz-in-garnet system, to avoid

significant pressure relaxation (>3%) in the bottom half of inclusion, the dimensionless depth needs to be above at least 1.5

(Fig. 4). To avoid significant pressure relaxation in the entire quartz inclusion, the dimensionless depth needs to be above ~2.

Therefore, we recommend a safe dimensionless depth of ~2 for quartz-in-garnet Raman-barometry (see also Mazzucchelli et

al., 2018).

For commonly used quartz-in-garnet Raman-barometry, our results show that below 550~600 $^{\circ}$C, the effect of viscous

relaxation can be negligible. Above ca. 650~750 $^{\circ}$C, the effect of viscous relaxation needs to be taken into account

depending on particular $P$-$T$ path, garnet composition and time scale (Fig. 5, Fig. 6). This is similar to the empirical estimate

ca. 750 $^{\circ}$C in Walters and Kohn (2014). It is also shown that the preserved residual pressure may even increase due to

viscous relaxation if viscous resetting occurs at peak $P$ condition (Fig. 8). This is simply because viscous creep does not only

relax the overpressure in quartz inclusion, but also the underpressure that develops along prograde $P$-$T$ path. Meanwhile, the

amount of viscous relaxation is time-dependent ($De$ is a function of the operating time of viscous relaxation). Thus, the

above temperature criterion for Raman-barometry applies only for exhumation lasting at million years' time scale. A higher

temperature criterion for Raman-barometry (e.g. ~1000 $^{\circ}$C for garnet host at high pressure close to coesite-quartz transition)

is applicable for more rapid exhumation, e.g. xenolith ascent carried by magma (Zhong et al., 2018b) or garnet synthesis

experiments that lasts hours/days (Thomas and Spear, 2018).

## 4.2 Implications to garnet overstepping

Quartz-in-garnet Raman-barometry has been used to determine the entrapment pressure, i.e. garnet nucleation/growth conditions and compared to the *P-T* conditions determined based on phase equilibria/classical chemical thermobarometry (Castro and Spear, 2017; Spear et al., 2014). As has been shown in Fig. 8, viscous resetting occurs when the inclusion-host system is brought to high temperature (>600~700 $^{o}$C). Even if the quartz inclusion is entrapped at lower *P-T* conditions, e.g. the garnet entrance conditions, the preserved residual inclusion pressure may still be significantly higher than predicted from the actual entrapment *P-T* conditions using pure elastic model. In this case, erroneous results may emerge if one uses the relaxed residual quartz inclusion pressure to determine the entrapment pressure. In case of significant viscous resetting at peak *T* conditions followed by decompression, as in the case of some HP rocks, apparent garnet growth overstepping will be inferred (see Fig. 8b). Care must thus be taken to interpret the discrepancy between the results of quartz-in-garnet Raman barometry and phase equilibria. As shown in the example with synthetic clockwise *P-T* path (Fig. 8), ca. 3~4 kbar apparent overstepping is yielded by considering viscous resetting at peak *T* condition. The amount of apparent overstepping will be even larger if the exhumation process happens faster (current model assumes 10 Ma decompression time).

## 5. Conclusions

We presented a 1D visco-elasto-plastic model to study the inclusion-host system undergoing prograde/retrograde *P-T* path. Nondimensionalization yields two controlling parameters, Deborah number (*De*) and cohesion number ($C^*$) that control the amount of viscous and plastic relaxation of the residual pressure of inclusion. Both *De* and $C^*$ must be higher than one to avoid relaxation due to viscous creep and plastic yield. A simplified analytical solution for inclusion pressure (Eq. 32) close to stress-free thin-section surface is derived. It is suggested that the distance between thin-section surface and inclusion must be higher than 2~3 times the inclusion radius to avoid stress relaxation.

## Code availability

The MATLAB code to reproduce the results of quartz-in-garnet system is uploaded with the paper.



**Author contribution**

X.Z. designed the numerical/analytical model and wrote the MATLAB code. All the co-authors contributed in discussion and wrote the manuscript together.

**Competing interests**

The authors declare no conflict of interest.

**Acknowledgements**

This work is supported by MADE-IN-EARTH ERC starting grant (n.335577) to LT and Swiss National Science Foundation (P2EZP2_172220) to XZ.

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



**Table**

**Table 1**. Averaged cohesion from microhardness tests for some minerals at room conditions. Cohesion is converted from microhardness based on $C_h = H/C_g$, where the geometry constant $C_g$ is taken as 3. Raw data are dependent on crystallographic orientation, composition and applied load that are examined in some of the involved references.

| Minerals | Cohesion (GPa) |
|---|---|
| calcite[2] | 0.6 |
| zircon[4] | 1.2 |
| dolomite[2] | 1.5 |
| orthoclase[1] | 2.3 |
| andalusite[1] | 2.3 |
| diopside[3] | 2.7 |
| sillimanite[1] | 3.7 |
| quartz[1] | 4.0 |
| kyanite[1] | 4.0 |
| spinel[5] | 4.1 |
| grossular[1] | 4.4 |
| almandine-pyrope[1] | 5.0 |

[1]Data reported in Whitney et al. (2007).

[2]Data reported in Wong and Bradt (1992). The reported data for calcite and dolomite are averaged from the applied load and azimuthal angle from $[10\bar{1}\bar{1}]$.

[3]Data reported in Smedskjaer et al. (2008).

[4]Data reported in Yuan et al. (2017)

[5]Data reported in Dekker and Rieck (1974). The reported data are averaged from the applied load at [110] and [100].






**Figures**

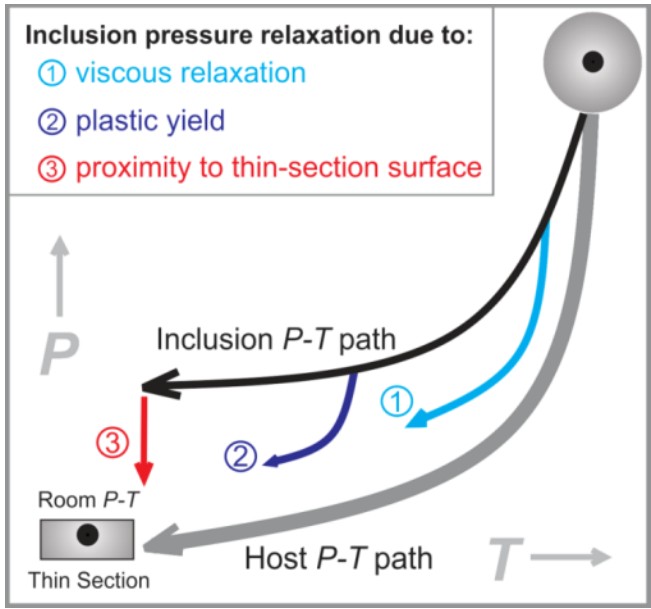

Fig. 1. Schematic illustration for the residual pressure development and relaxation. The grey and black curves are retrograde

*P-T* paths for host and inclusion, respectively. Pressure relaxation is possibly due to following reasons: 1) viscous relaxation

preferentially occurs at high temperature conditions; 2) plastic yield commonly occurs at low confining pressures where

residual pressure is high; 3) thin-section preparation that drives inclusion close to thin-section surface.


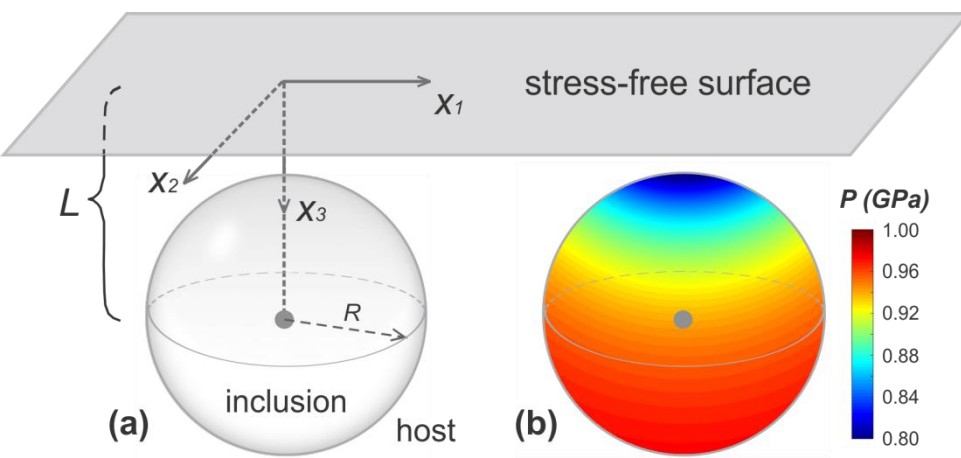


Fig. 2. **A:** Model configuration of mineral inclusion close to thin-section surface. The distance between the surface to inclusion centre is denoted by $L$. **B:** Pressure distribution on $x$-$z$ plane ($L = 1.5R$). Initially the inclusion contains 1GPa residual pressure and is relaxed when brought next to the stress-free surface. The analytical solution of Eq. 32 is used for the pressure plot.






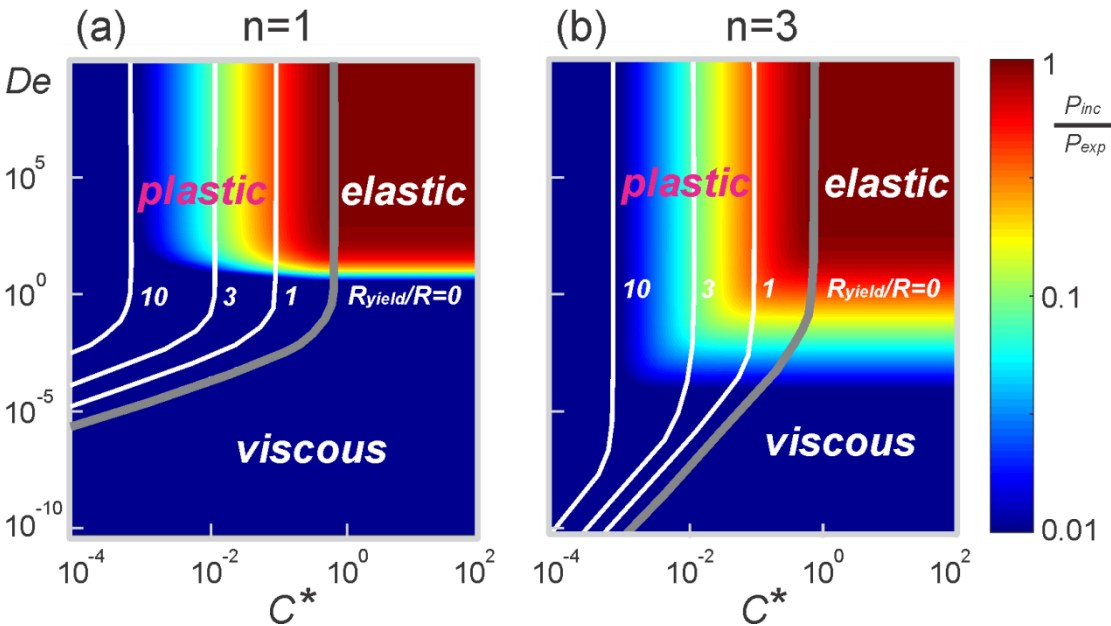

Fig. 3. Inclusion pressure as a function of Deborah number and Cohesion number given different stress exponents. The contours denote the radius of plastic yield region $R_{yield}$ scaled by inclusion radius. The thick grey contour represents the onset of plastic yield. Three regimes are labelled: 1) elastic ($De > 1$, $C^* > 1$); 2) viscous ($De < 1$ and $C^*$ is above the onset of plastic yield); 3) plastic ($C^* < 1$, $De$ is above the onset of plastic yield). To obtain the results, a residual pressure is prescribed at the beginning and the confining pressure and temperature are fixed, i.e. no temporal variations of $P$-$T$ conditions.





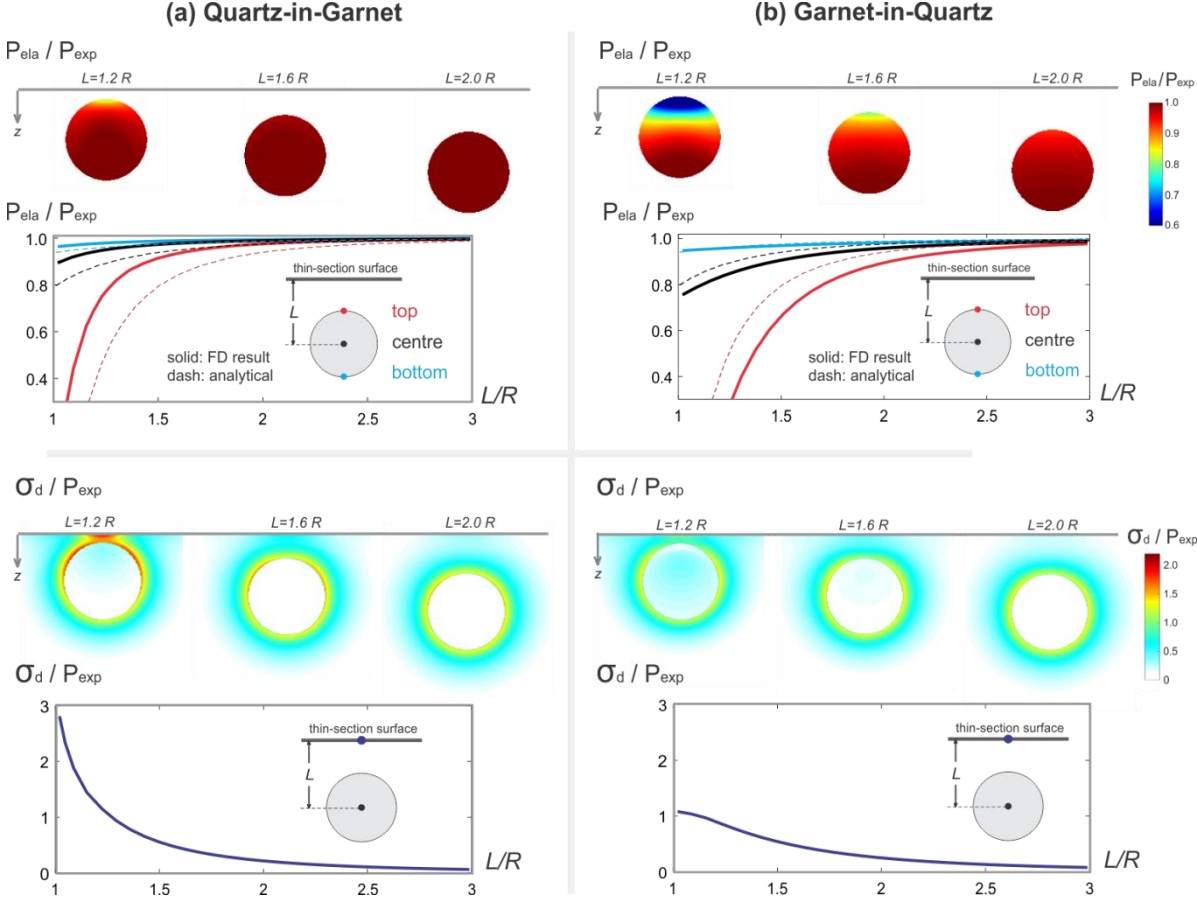

Fig. 4. Dimensionless pressure and differential stress plotted on *x-z* plane, or as a function of dimensionless depth. **A:** Quartz-in-pyrope system; **B**: Pyrope-in-quartz system. For the profiles, pressure and differential stress are measured at different locations denoted by the coloured dots. In the top panel, the dashed curves in the pressure plot are based on the analytical solution in Eq. 32 considering the same elastic moduli between inclusion and host, while the solid curves are based on finite difference results. The discrepancy between the solid (numerical solution) and dashed (analytical solution)

curves in **A** is due to the fact that the host elasticity is different than the inclusion.



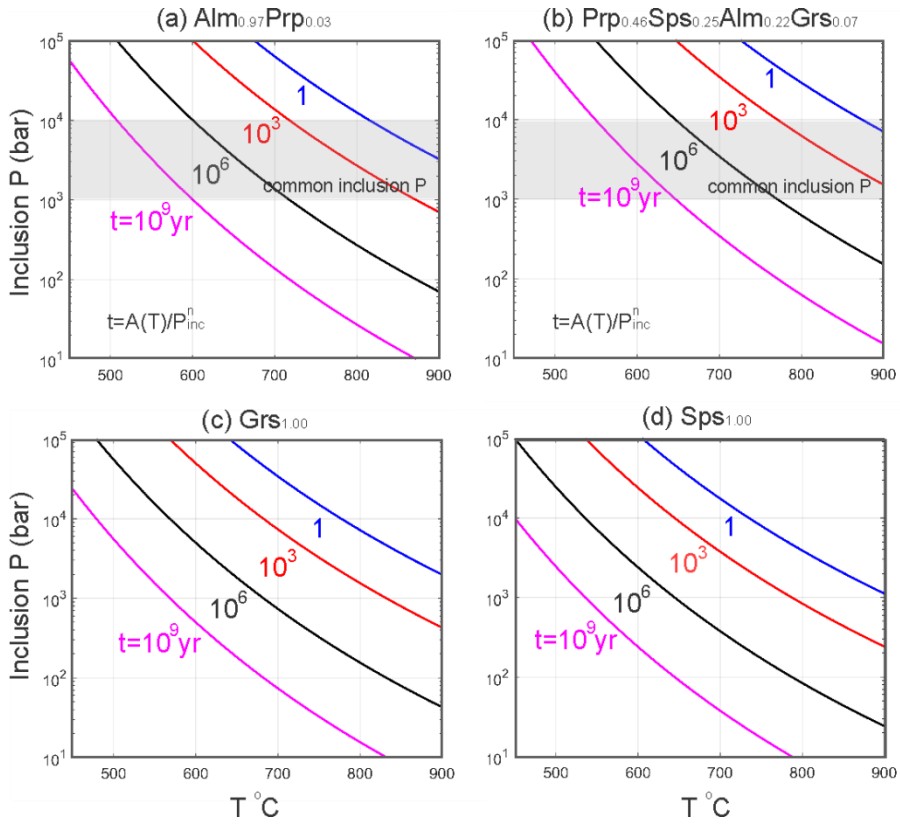

Fig. 5. Viscous relaxation time (in years) of different garnet host as functions of temperature and inclusion overpressure. The viscous relaxation time is calculated based on the expression of Deborah number ($De$=1) in Eq. 18. The viscous pre-factor ($A$) is $T$ dependent and is obtained using the flow law from Wang and Ji (1999). The melting temperature is from Karato et al. (1995) (the melting temperature of almost pure almandine is taken from the data of Mohawk Garnet Inc. to be 1588K). Shear modulus is from Bass (1995). Viscosity pre-factor $A$ is calculated as: $\frac{G^n}{2B}\exp(\frac{g \cdot T_m}{T})$, where $B = \exp(40.1)$ in $s^{-1}$, $g = 32$ and the stress exponent $n$=3 (Wang and Ji, 1999). The four garnet endmembers are almandine (Alm), grossular (Grs), pyrope (Prp) and spessartine (Sps).

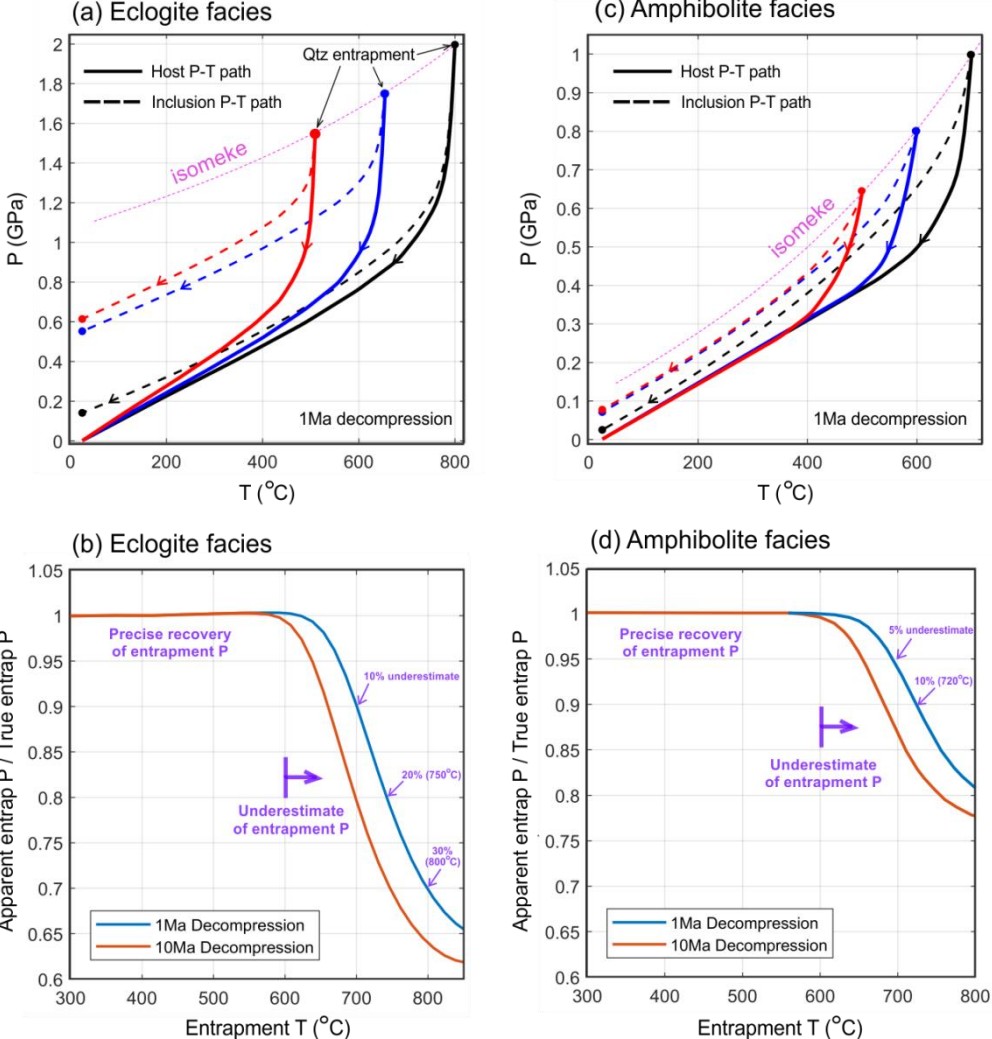


Fig. 6. **A**. Synthetic retrograde *P-T* paths from eclogite facies metamorphic conditions. The quartz inclusions are entrapped

within almandine at different peak *P-T* conditions along the same isomeke. Due to viscous relaxation, the residual *P* is lower

than the pressure predicted by an elastic model. In **B**, the apparent entrapment *P* is calculated based on the relaxed residual

inclusion pressure given different entrapment *T* along the elastic isomeke that is given in **A**. Pressure relaxation is manifested

by lower values of apparent entrapment *P* and it becomes more significant if the host experiences high temperatures with

time. **C** and **D** are the same plots for amphibolite-facies entrapment conditions. The amount of viscous relaxation is less

compared to eclogite facies due to the lower magnitude of inclusion overpressure and the stress dependent viscosity of garnet

host. Pure almandine garnet is used as host and its flow law is from Wang and Ji (1999).

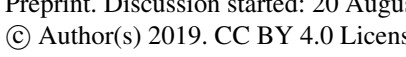



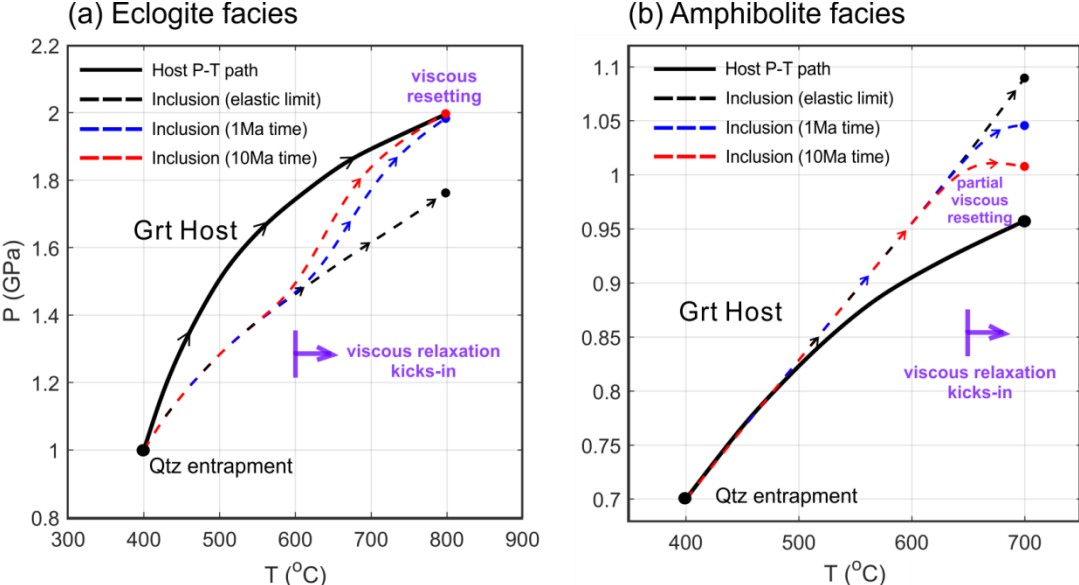

Fig. 7. Prograde *P-T* path for inclusion (dashed curve) and host (solid curve). **A** is for rocks that experienced eclogite-facies

peak *P-T* conditions. The quartz inclusion is entrapped at 400 °C and 1 GPa. Along the given prograde *P-T* path, viscous

relaxation becomes significant at >600 °C. The duration of prograde *P-T* path is illustrated with different colour (1 Ma and

10 Ma, see legend). At 800 °C, the quartz inclusion pressure is reset to the confining pressure (host). For rocks that

experienced amphibolite-facies peak *P-T* conditions, viscous relaxation becomes significant at ca. 650~700 °C and the quartz

inclusion pressure is partially reset at 700 °C. Pure almandine garnet is used as host and its flow law is from Wang and Ji

(1999).



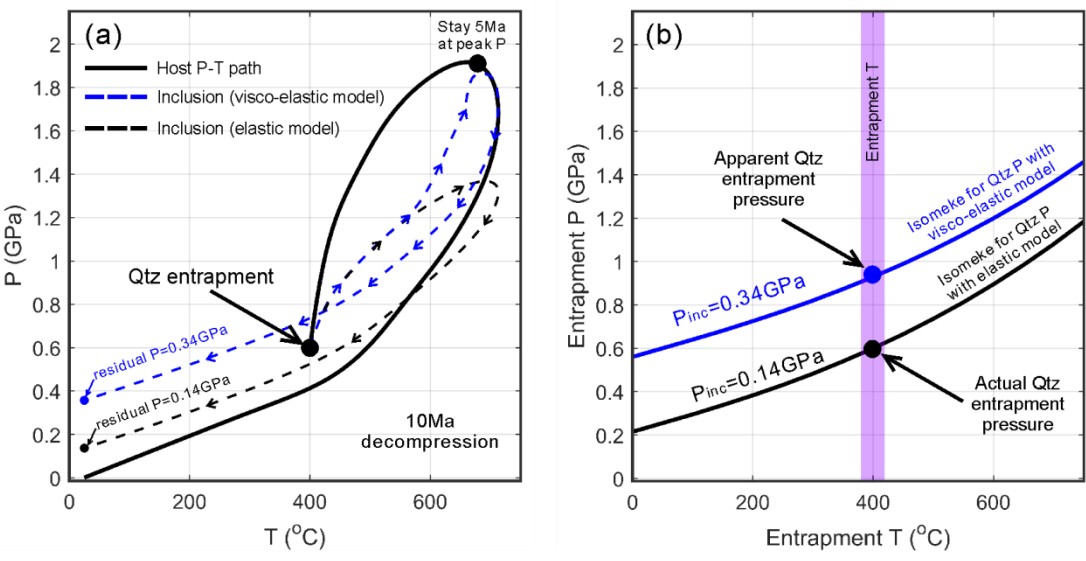

Fig. 8. **A**. Clockwise *P-T* path of inclusion (dashed curve) and host (solid curve). The dashed black curve shows the inclusion *P-T* path based on pure elastic model and the blue dashed curve is based on visco-elastic model. The quartz inclusion is entrapped into almandine garnet at 400 °C, 0.6 GPa. The prograde *P-T* path lasts 5 Ma, and the rock stays at peak *P* for 5 Ma before retrograde *P-T* path, which lasts 10 Ma. The residual pressure preserved by the quartz inclusion that was subject to viscous relaxation is in fact higher than the elastic limit. Therefore, its apparent entrapment pressure calculated using elastic isomeke becomes higher than the actual entrapment pressure as shown in **B**. This may lead to ca. 3~4 kbar apparent overstepping effect. The almandine flow law is from Wang and Ji (1999).