# Peer review of "Post-entrapment modification of residual inclusion pressure and its implications to Raman elastic"

_Solid Earth, 2019_

## Referee Comment (RC1) · Viktoriya Yarushina (Referee) · 24 Sep 2019

Presented manuscript discusses pressure variations around inclusions in the homogeneous rock matrix and their implications for the accuracy of Raman-thermobarometry measurements. Authors study two different processes that might lead to stress changes around a single inclusion: stress relaxation on a geological time scale due to viscoplastic stress relaxation and proximity of the free surface to the mineral inclusion during sample preparation in the lab. Authors show that both stress relaxation and presence of free surface might alter stresses inside inclusion and in the host matrix. Hence, they might lead to erroneous estimations during Raman-thermobarometry. While this is an interesting paper, its logic and presentation could be improved. Authors are using two different problem setups and switch in the text from one of them

to another without much mentioning of it. I would recommend revising the manuscript and clearly separate results related to sample preparation (i.e., setup with elastic solution for half-space) and results related to stress relaxation on a geological time scale. Some other points to consider:

- Throughout the manuscript. Use of term "relaxation" with respect to changes in stress due to the presence of free boundary as in sample preparation setup and due to plastic effects is incorrect and confusing. See e.g. original paper by Zhang (1998) where he states that "Plastic yielding does not relax the stress but does limit the deviatoric stress" (page 215). I would call these processes rather "stress release". Besides, authors consider viscoelastoplastic model for stress relaxation where plasticity contributes simultaneously with viscosity, i.e. purely plastic (or elastoplastic) stress release is not considered.

- Lines 48-50. Authors state that "Mechanical models show that both viscous creep (dislocation or diffusion creep of host) and plastic yield (radial or tangential microcracks) can cause significant pressure relaxation (Dabrowski et al., 2015; Zhang, 1998)." While cited references indeed present viscous relaxation models, none of them presents mechanical model that shows plastic yield or radial and tangential microcracks. Care with references is needed.

- Section 2.1. Logic of this section can be improved. Equation (2) uses results of equation (4), which is further in the text. It is better to introduce plastic flow law (4) first and then give summary equation for total strain rate (2).

- Lines 84 and 86. The choice of references for classical Tresca yield criterion and associated plastic flow law is a bit odd. There are much older, standard and very good textbooks that introduced those, e.g. [Hill, 1950; Kachanov, 1971].

- Lines 85 – 89 and throughout the text. Parameter C in the Tresca yield condition is not a cohesion. It is a half of the yield limit for simple tension of the host matrix in the case of spherical inclusions [Hill, 1950; Kachanov, 1971]. This is important to

note because later in the text (in the discussion section) authors make estimations of this parameter based on the experimental data for cohesion and make conclusions for Raman-thermobarometry. Also, I would want to mention that Tresca criterion represents yielding due to dislocation sliding in crystalline materials at high pressures and thus cannot be taken as a proxy for fracturing. You need a more thorough discussion on the deformation mechanisms in the host rock to justify the choice of yield function and viscosity (eqn (5)). There are various deformation mechanism maps for viscosity can be found in the literature.

- Line 83. Wrong statement: "lambda is the plastic multiplier (s-1) which guarantees that the plastic yield criterion is not exceeded". Plastic multiplier provides the amount of the plastic deformation. However, in the numerical codes it is indeed calculated from yield criterion and consistency conditions.

- Equation (9). Explain parameter delta.

- Lines 102-105. Authors write that "This is done by choosing the following independent scales: the inclusion radius, temperature change, time, viscosity pre-factor of host, plastic cohesion of host, and the expected pressure perturbation that is given as follows..." Again, be careful with your statements. These scales are not independent. You can have only one independent scale of pressure, time, temperature, length. Thus, viscosity factor and cohesion would be already dependent parameters.

- Line 109. It is not entirely clear why Pexp is chosen as a scale. Please explain this parameter, where it comes from. Also check eqn (11). It is not logical to scale yield function F and cohesion to different scales.

- Lines 123-127. Again, C is not cohesion. Please rewrite this para. Statements on the range of viscosity and yield limit must be supplemented with references and even better if with realistic numbers. I doubt that one can expect orders of magnitude variation in yield strength of minerals as it is stated by authors: "the cohesion of difference mineral may also vary by many several factors, potentially orders of magnitude".
- Section 2.3. Given that there is no reference for the numerical approach, I assume it is original. Or was it previously reported somewhere? In general, there is a different level of detail in the paper. E.g. there is too much focus on standard non-dimensional analysis and almost nothing on non-trivial viscoelastoplastic numerical scheme or on the elastic numerical solution for half-space in the following sections.

- Section 2.4. It is hard to see the value of this section in the paper. Authors present analytical solution previously derived by Seo and Mura [1979]. However, no meaningful analysis or conclusions for the topic of the paper (i.e. Raman-thermobarometry) were derived from this solution. Its use for benchmarking of numerical code is also limited as analytical and numerical results are different due to various assumptions about material properties of inclusion. On the other hand, other analytical solutions used for benchmarking in section 2.3 are not resented at all. If authors choose to keep this solution, I would recommend discussing carefully boundary, initial conditions and its relation to the Raman-thermobarometry. For example, is it an incremental solution and does it show changes in stress from a specific initial condition? Or does it show stress distribution in an inclusion-host system without initial pre-stress? What do we learn from this solution?

- Line 172. Why Pexp is referred to as "initial residual pressure"? As this solution is presented now, there is no process in it, only static force equilibrium.

- Section 3.1. Switch from one problem setup to another comes very abrupt here. Please document your simulation setup (geometry, boundary and initial conditions, properties of the host and inclusion) and state which problem you address (i.e. stress relaxation or sample preparation). The title of this section is inconsistent with the following sections.

- Line 183. "This diagram may assist petrological investigations because De and C* can be evaluated based on experimental rock deformation data for different minerals..." Please discuss how De and C* can be evaluated based on experimental data. Which

data is available?

- Line 196. Awkward phrase: "...and De is located above the plastic onset..." Please reformulate.

- Section 3.2. Describe problem setup, boundary and initial conditions. Given that you have two different problem setups in the paper it is confusing. Governing equations and a little bit info about numerical implementation would also fit here rather than supplementary materials in the same way as you present another model. Without such descriptions, it is hard to see the relevance to sample preparation problem. Do you consider just an equilibrium stress state, or do you have an incremental formulation that considers initial condition? Check for use of word "relaxation" here and rather use "release". Check also for consistent use of "quartz-in-almandine" and "quartz-in-garnet" terms.

- Line 210-216. What are the implications for sample preparation, e.g. in terms of thickness, etc?

- Line 223. "Assuming that the thin-section surface is sufficiently far away from a quartz inclusion and no microcracks appear around quartz inclusion..." I recommend replacing "microcracks" with "yield" as your solution does not consider microcracks and there is a discussion on microfractures later.

- Line 227. "The flow law of garnet from Wang and Ji (1999) is applied..." Please describe briefly this law.

- Line 272. "The three mechanisms investigated here, i.e. viscous creep, plastic yield and proximity of inclusion..." Plastic yield was studied only together with creep, i.e. on a geological time scale. Plastic yield without creep as might occur e.g. during sample preparation was not studied. Thus, I think it is more appropriate here to use term viscoplastic flow instead of plastic yield.

- Section 4.1. C is not cohesion, please check relevant values and your estimations for

Ch.

- Lines 283-289. "This suggests that plastic yield does not occur in an idealized scenario of isotropic, spherical quartz inclusion entrapped in infinite garnet host. However, such an ideal scenario is highly improbable in natural samples. The observed cracks in garnet host may be formed due to potential reasons including: 1) elevated differential stress when the inclusion is close to thin-section surface ("ring" shaped pattern in Fig. 4a); 2) stress concentration at the corners of quartz inclusion (Whitney et al., 2000); 3) anisotropic elastic deformation of the quartz inclusion (e.g. Murri et al., 2018); 4) pre-fractures/weakness in garnet host before the entrapment of quartz inclusions."

While I agree with the possibility of elevated differential stress and stress concentrations at the corners, I would like to emphasize that this statement is based on the solution for materials obeying Tresca criterion, which does not describe fracturing. To make conclusions about fractures around inclusion, one needs to consider other failure criteria such as Griffith or Mohr-Coulomb, where cohesion and tensile strength play major role. Solutions for plasticity onset and failure pattern in elastoplastic and viscoplastic rocks with these failure criteria are available in the literature. They would give other estimations for pressure necessary to induce fractures.

- Conclusions. "We presented a 1D visco-elasto-plastic model to study the inclusion-host system undergoing prograde/retrograde P-T path" There are at least two different models in this paper.

- "A simplified analytical solution for inclusion pressure (Eq. 32) close to stress-free thin-section surface is derived." The solution presented by authors was not new, it was reroduced after original aer by Seo and Mura [1979].

- Please also make some statements on the implications for Raman-thermobarometry and how to use your results.

References:

Hill, R. (1950), The mathematical theory of plasticity, 356 pp., Clarendon Press, Oxford,. Kachanov, L. M. (1971), Foundations of the theory of plasticity, xiii, 482 p. with illus. pp., North-Holland Pub. Co., Amsterdam,.

Please also note the supplement to this comment:
https://www.solid-earth-discuss.net/se-2019-124/se-2019-124-RC1-supplement.pdf

---

## Referee Comment (RC2) · Anonymous Referee #2 · 5 Nov 2019

I apologize for the delay of the review process but I had to many administrative issues to deal with first. The manuscript is actually one of those that make the review process very easy. It is well-written, it has we—organized structure and the addresses a timely topic with a novel approach. On top what referee #1 has been already mentioned, I only have suggestions, basically no real criticism.

I like the way the authors explain the methods in chapter 2. And finally discuss the most important findings of the study in a reasonable detail. I would only suggest to spent less words on the "distance to surface" issue (chapter 3.2) as this is not really new, but focusing more on the "over-" and "underestimation" of the pressure depending on the PT paths the samples take. This is really exciting and should be highlighted even more.

[Figure]

To me the authors could consider to keep the inclusion-host relationship a bit broader in the begin of the introduction, such as considering inclusions in a bit broader context (see Farber et al 2014 CMP, for instance), but this may result in a less sharp structure.

Kind regards

———————————————

---

## Author Response (AR1)

[revised manuscript text omitted]

where $Z = \frac{\cancel{\Delta t G}}{\cancel{\Delta t G + \eta}} \frac{\Delta t G}{\Delta t G + \eta}$ is the viscoelastic coefficient, $\tau_{rr}^o$ is the radial component of deviatoric stress in the previous time step, $P^o$ is the pressure in previous time step. If the yield criterion in Eq. $\underline{5}\cancel{3}$ is exceeded ($F > 0$), the plastic multiplier must be adjusted to drive $F$ to zero. This can be achieved by substituting the deviatoric stress (Eq. $\underline{8}\cancel{7}$) into Eq. $\underline{5}\cancel{3}$ and let $F = 0$. Therefore, we obtain $\lambda$ as follows:

$$\lambda = \delta \dot{e}_{rr} + \frac{(1-Z)sign(\tau_{rr})}{2\eta Z}\cancel{\tau_{rr}^o} - \frac{\cancel{C}}{\cancel{3\eta Z}}\frac{(1-Z)\delta}{2\eta Z}\tau_{rr}^o - \frac{\delta C}{3\eta Z}, \quad \text{if } F > 0 \text{ (otherwise } \lambda = 0\text{).} \tag{$\underline{10}\cancel{9}$}$$

**2.2 Nondimensionalization**

The variables in the above equations can be scaled to derive nondimensional parameters for better understanding the behaviour of the inclusion-host system. This is done by choosing the following six parameters to nondimensionalize the system of equations: the temperature drop of the host-inclusion system $\Delta T$, the inclusion radius $R$,  the time of the $P - T$ path $t^*$, the host's viscosity pre-factor $\cancel{A_h \text{ of}}A_\text{h}$, the host,'s plastic yield strength $\cancel{C_h \text{ of host,}}C_\text{h}$, and the expected pressure perturbation $\cancel{P_{exp}}P_\text{exp}$ that is given as follows:

$$\cancel{P_{exp} = \frac{\Delta P(\beta_i - \beta_h) - \Delta T(\alpha_i - \alpha_h)}{\beta_i + 3/4 G_h}}; P_\text{exp} = \frac{\Delta P(\beta_i - \beta_h) - \Delta T(\alpha_i - \alpha_h)}{\beta_i + 3/4 G_h}, \tag{$\underline{11}\cancel{10}$}$$

The involved physical variables are scaled as shown below:

where $\Delta P, \Delta T$ are the confining pressure and temperature drops from entrapment to the Earth's surface, $\beta_i$ and $\beta_h$ are the compressibility of inclusion and host, $\alpha_i$ and $\alpha_h$ are the thermal expansion coefficients of inclusion and host, $G_h$ is the shear modulus of host. The number $P_{\text{exp}}$ is the expected amount of residual inclusion pressure after exhumation assuming linear thermo-elasticity and infinite host (Zhang, 1998). It is noted that this is not the actual final residual inclusion pressure, but merely a scale to nondimensionalize the stress (and pressure). By choosing $P_{\text{exp}}$ as the stress scale, the inclusion residual pressure is expected to be between 0 and 1 for a case of cooling and decompression. This pressure scale allows convenient quantification for the amount of pressure modification due to viscous creep and plastic yield. The involved physical variables are scaled as shown below:

$$r = R\,\bar{r}$$

$$\beta = \frac{1}{P_{\text{exp}}}\frac{1}{P_{\text{exp}}}\bar{\beta}$$

$$G = P_{\text{exp}}P_{\text{exp}}\bar{G}$$

$$\alpha = \frac{1}{\Delta T}\bar{\alpha}$$

$$P = P_{\text{exp}}P_{\text{exp}}\bar{P}$$

$$\dot{T} = \frac{\Delta T}{t^*} = \frac{\Delta T}{t^*}\bar{\dot{T}}$$

$$\tau_{rr} = P_{\text{exp}}P_{\text{exp}}\overline{\tau_{rr}}$$

$$C = G_h C_h \bar{C}$$

$$\eta = P_{\text{exp}}P_{\text{exp}}t^*\bar{\eta}$$

$$F = P_{\text{exp}}C_h\bar{F}$$

$$\Delta t = t^*\overline{\Delta t}$$

$(12)$

$$A = \bcancel{A_{\bar{h}}} A_{\mathrm{h}} \bar{A}$$

$$\lambda = \frac{1}{t^*} \bar{\lambda}$$

$$v_r = \frac{R}{t^*} \bar{v}_r$$

where the overhead bars indicate dimensionless properties. Substituting these scaling equations into Eq. 1, 9 and 8, we get:

$$\frac{\partial \overline{\tau_{rr}}}{\partial \bar{r}} + \frac{3\overline{\tau_{rr}}}{\bar{r}} - \frac{\partial \bar{P}}{\partial \bar{r}} = 0, \tag{13\,12}$$

$$\bar{P} = \overline{P^o} + \frac{1}{\bar{\beta}}\left[\bcancel{-\overline{\Delta t}\frac{\partial \bar{r}^2 \overline{v_r}}{\bar{r}^2 \partial \bar{r}} + \bar{\alpha}\bar{T}}\right]\left[-\overline{\Delta t}\frac{\partial \bar{r}^2 \overline{v_r}}{\bar{r}^2 \partial \bar{r}} + \bar{\alpha}\bar{T}\right], \tag{14\,13}$$

$$\overline{\tau_{rr}} = \frac{4}{3}\bar{\eta}\bar{Z}\left(\frac{\partial \bar{v}_r}{\partial \bar{r}} - \frac{\bar{v}_r}{\bar{r}}\right) + (1 - \bar{Z})\overline{\tau_{rr}^o} - 2\bar{\eta}\bar{\lambda}\delta\bar{Z}, \tag{15\,14}$$

where dimensionless viscosity, viscoelastic coefficient and plastic multiplier are expressed as:

$$\bar{\eta} = De \cdot \bar{A}|\overline{\tau_{rr}}|^{1-n}, \tag{16\,15}$$

$$\bar{Z} = \bcancel{\frac{\overline{\Delta t}\bar{G}}{\overline{\Delta t}\bar{G} + \bar{\eta}}} = \frac{\overline{\Delta t}\bar{G}}{\overline{\Delta t}\bar{G} + \bar{\eta}}, \tag{17\,16}$$

$$\bar{\lambda} = \frac{4}{3}\delta\left(\frac{\partial \bar{v}_r}{\partial \bar{r}} - \frac{\bar{v}_r}{\bar{r}}\right) + \frac{(1-\bar{Z})\delta}{2\bar{\eta}\bar{Z}}\overline{\tau_{rr}^o} - C^* \cdot \bcancel{\frac{\bar{C}}{3\bar{\eta}\bar{Z}}}\frac{\bar{C}}{3\bar{\eta}\bar{Z}}, \text{ if } \frac{3}{2}\delta\overline{\tau_{rr}} - \bcancel{C^* \cdot \bar{C}}\bar{C}\bar{F} > 0. \tag{18\,17}$$

Two dominant dimensionless numbers emerge after nondimensionalization. They are Deborah number $De$ and dimensionless yield strength $C^*$ defined as follows:

$$De = \bcancel{\frac{A_{\bar{h}}/P_{exp}^n}{t^*}}\frac{A_{\mathrm{h}}/P_{\exp}^n}{t^*}, \tag{19\,18}$$

$$C^* = \bcancel{\frac{C_{\bar{h}}}{P_{exp}}}\frac{C_{\mathrm{h}}}{P_{\exp}}. \tag{20\,19}$$

The

[revised manuscript text omitted]

*General reply to reviewer 1*

*We thank the reviewer for the very careful examination of our work and helpful comments that have greatly improved the quality of our paper. All the comments from the reviewer have been carefully considered and point-by-point replies have been provided in this letter. The original comments from the reviewer are in "*regular black*" and our replies are in "italic blue".*

Comments from Reviewer 1

Presented manuscript discusses pressure variations around inclusions in the homogeneous rock matrix and their implications for the accuracy of Raman-thermobarometry measurements. Authors study two different processes that might lead to stress changes around a single inclusion: stress relaxation on a geological time scale due to visco-plastic stress relaxation and proximity of the free surface to the mineral inclusion during sample preparation in the lab. Authors show that both stress relaxation and presence of free surface might alter stresses inside inclusion and in the host matrix. Hence, they might lead to erroneous estimations during Raman-thermobarometry.

While this is an interesting paper, its logic and presentation could be improved. Authors are using two different problem setups and switch in the text from one of them to another without much mentioning of it. I would recommend revising the manuscript and clearly separate results related to sample preparation (i.e., setup with elastic solution for half-space) and results related to stress relaxation on a geological time scale.

*This is a very helpful comment. Indeed, our previous structure mix the two main parts of the model.*

*We have re-structured the entire manuscript to clearly separate the two different models and corresponding results/discussions. We hope our revised structure will be clearer for readers.*

Throughout the manuscript, use of term "relaxation" with respect to changes in stress due to the presence of free boundary as in sample preparation setup and due to plastic effects is incorrect and confusing. See e.g. original paper by Zhang (1998) where he states that "Plastic yielding does not relax the stress but does limit the deviatoric stress". I would call these processes rather "stress release". Besides, authors consider visco-elasto-plastic model for stress relaxation where plasticity contributes simultaneously with viscosity, i.e. purely plastic (or elastoplastic) stress release is not considered.

*We agree with the reviewer's comment and we have followed the reviewer's suggestion to change "stress relaxation" in plastic yield situation and proximity to surface situation into "stress release".*

*We also added new paragraphs (in "Introduction" and "Residual pressure affected by viscous/plastic flow" sections) to make clarification that in our case, we refer to viscous relaxation as a time-dependent process and plastic yield as time-independent process that limits the deviatoric stress in the host, thus also limits the residual inclusion pressure. Both of them are irreversible.*

Lines 48-50. Authors state that "Mechanical models show that both viscous creep (dislocation or diffusion creep of host) and plastic yield (radial or tangential micro-cracks) can cause significant

pressure relaxation (Dabrowski et al., 2015; Zhang, 1998)." While cited references indeed present viscous relaxation models, none of them presents mechanical model that shows plastic yield or radial and tangential microcracks. Care with references is needed.

*We agree that in Dabrowski et al. 2015, plastic yielding was not studied. But in Zhang 1998, he showed the effect of plastic yield on the stress distribution (Fig. 2B). Due to the limited differential stress in the host upon plastic yield, the stress state in the inclusion will be changed. In his section 4.3, Zhang (1998) focused on the radial cracks formed during decompression. Although he did not quantitatively show the amount of pressure release but he did mention about the fractures formed around various mineral inclusions depending on the bulk modulus of the inclusion and host.*

Section 2.1. Logic of this section can be improved. Equation (2) uses results of equation (4), which is further in the text. It is better to introduce plastic flow law (4) first and then give summary equation for total strain rate (2).

*This part is now clarified. We thank the reviewer for pointing it out.*

Lines 84 and 86. The choice of references for classical Tresca yield criterion and associated plastic flow law is a bit odd. There are much older, standard and very good textbooks that introduced those, e.g. [Hill, 1950; Kachanov, 1971].

*Reference is updated, we added Kachanov (1971) as suggested.*

Lines 85 – 89 and throughout the text. Parameter C in the Tresca yield condition is not cohesion. It is a half of the yield limit for simple tension of the host matrix in the case of spherical inclusions [Hill, 1950; Kachanov, 1971]. This is important to note because later in the text (in the discussion section) authors make estimations of this parameter based on the experimental data for cohesion and make conclusions for Raman-thermobarometry. Also, I would want to mention that Tresca criterion represents yielding due to dislocation sliding in crystalline materials at high pressures and thus cannot be taken as a proxy for fracturing. You need a more thorough discussion on the deformation mechanisms in the host rock to justify the choice of yield function and viscosity (Eq. 5). There are various deformation mechanism maps for viscosity can be found in the literature.

*This is a valid point and we agree with the reviewer. We have updated the reference based on the reviewer's suggestion and carefully checked the entire manuscript to adjust "cohesion" into "yield strength". We have also removed the parts for "fracturing" and replace them by plastic yield.*

Line 83. Wrong statement: "lambda is the plastic multiplier ($s^{-1}$) which guarantees that the plastic yield criterion is not exceeded". Plastic multiplier provides the amount of the plastic deformation. However, in the numerical codes it is indeed calculated from yield criterion and consistency conditions.

*This sentence has been adjusted as suggested.*

Equation (9). Explain parameter delta.

*Clarified as suggested.*

Lines 102-105. Authors write that "This is done by choosing the following independent scales: the inclusion radius, temperature change, time, viscosity pre-factor of host, plastic cohesion of host, and the expected pressure perturbation that is given as follows" Again, be careful with your statements. These scales are not independent. You can have only one independent scale of pressure, time, temperature, length. Thus, viscosity factor and cohesion would be already dependent parameters.

*This is correct. We thank the reviewer for this point. We have corrected this sentence. The parameters we used to nondimensionalize the system are not independent, which has been noted in the revised manuscript by deleting "independent".*

Line 109. It is not entirely clear why Pexp is chosen as a scale. Please explain this parameter, where it comes from.

*This is a good point. We missed the explanation for Pexp. It comes from the paper of Zhang (1998) which described the amount of residual pressure assuming constant bulk modulus and thermal expansivity. Using it as a pressure scale is advantageous as the inclusion pressure will vary from 0 to 1. Now this is clarified in the text.*

Also check Eq. 11. It is not logical to scale yield function F and cohesion to different scales.

*We thank the reviewer for this point. We have changed the scale for F with the yield strength of the host. This does not influence the other equations as F does not appear in other equations.*

Lines 123-127. Again, C is not cohesion. Please rewrite this para. Statements on the range of viscosity and yield limit must be supplemented with references and even better if with realistic numbers. I doubt that one can expect orders of magnitude variation in yield strength of minerals as it is stated by authors: "the cohesion of difference mineral may also vary by many several factors, potentially orders of magnitude".

*We have removed all "cohesion" words following the suggestion. We have removed the statement of "several orders of magnitude".*

Section 2.3. Given that there is no reference for the numerical approach, I assume it is original. Or was it previously reported somewhere? In general, there is a different level of detail in the paper. E.g. there is too much focus on standard non-dimensional analysis and almost nothing on non-trivial visco-elasto-plastic numerical scheme or on the elastic numerical solution for half-space in the following sections.

*We agree with the reviewer that the numerical approach is relatively short compared to other parts. We have added some new descriptions and references on this part. We would not say the numerical approach we presented as original, where we use finite difference method (staggered grid stencil) and iteratively solve for the three unknowns from three equations. The numerical stencil (staggered grid finite difference, mostly in 2D/3D) has been used widely (e.g. Gerya 2010, now cited in the text) and the iterative solver (Picard iteration) is also relatively common in numerical modelling.*

Section 2.4. It is hard to see the value of this section in the paper. Authors present analytical solution previously derived by Seo and Mura [1979]. However, no meaningful analysis or conclusions for the topic of the paper (i.e. Raman-thermobarometry) were derived from this solution. Its use for benchmarking of numerical code is also limited as analytical and numerical results are different due to various assumptions about material properties of inclusion. On the other hand, other analytical solutions used for benchmarking in section 2.3 are not resented at all. If authors choose to keep this solution, I would recommend discussing carefully boundary, initial conditions and its relation to the Raman-thermobarometry. For example, is it an incremental solution and does it show changes in stress from a specific initial condition? Or does it show stress distribution in an inclusion-host system without initial pre-stress? What do we learn from this solution?

*This is a very valid comment. A similar point has also been raised by the second reviewer. We have now reduced the focus of the "half-space" problem by moving all the derivations into appendix and we only left the final analytical formula for pressure distribution in the main text (which is a new result). We have added a new section discussing the application of the presented formula together with an updated figure (Fig. 3 in the revised version) to show the results of the analytical formula for pressure distribution of inclusion in half-space.*

*We agree that the application for numerical benchmark is limited by the fact that the presented formula is only applicable to homogeneous inclusion. However, due to the simplicity (compared to the analytical formulas of e.g Mindlin, and Seo & Mura) we argue that it is still useful. This we have discussed it in a new paragraph for a benchmarking purpose in the appendix.*

Line 172. Why Pexp is referred to as "initial residual pressure"? As this solution is presented now, there is no process in it, only static force equilibrium.

*Pexp is the initial overpressure that is already under force equilibrium in infinite host. As the inclusion approaches the thin-section surface, the inclusion pressure gets smaller than Pexp. This is now explained at the beginning of the new section of "Inclusion pressure modification due to proximity to thin-section surface".*

Section 3.1. Switch from one problem setup to another comes very abrupt here. Please document your simulation setup (geometry, boundary and initial conditions, properties of the host and inclusion) and state which problem you address (i.e. stress relaxation or sample preparation). The title of this section is inconsistent with the following sections.

*We have restructured the whole manuscript and clearly separate the results of the presented two models. Now this section directly continue from the visco-elasto-plastic 1D model so there should be no confusion on which model we refer to. We also added a short paragraph before it to clarify the simulation setup.*

Line 183. "This diagram may assist petrological investigations because De and C* can be evaluated based on experimental rock deformation data for different minerals..."Please discuss how De and C* can be evaluated based on experimental data. Which data is available?

*We have added new sentences in the following text to clarify it, e.g. using flow law to evaluate De and using microhardness test data to evaluate C*.*

Line 196. Awkward phrase: "...and De is located above the plastic onset..." Please reformulate.

*Done. We have changed it to De above one to describe the plastic dominant regime.*

Section 3.2. Describe problem setup, boundary and initial conditions. Given that you have two different problem setups in the paper it is confusing. Governing equations and a little bit info about numerical implementation would also fit here rather than supplementary materials in the same way as you present another model. Without such descriptions, it is hard to see the relevance to sample preparation problem. Do you consider just an equilibrium stress state, or do you have an incremental formulation that considers initial condition?

*This is a valid point. We have restructured the whole manuscript to split the two models. Also we have added new text here to elaborate the model setup. This we have clarified in new section 4.2 (place has been changed due to restructuring).*

Check for use of word "relaxation" here and rather use "release". Check also for consistent use of "quartz-in-almandine" and "quartz-in-garnet" terms.

*This has been pointed out before by the reviewer and we completely agree. We have checked the word "relaxation" in the entire manuscript to carefully split it with "release" due to different rheology.*

Line 210-216. What are the implications for sample preparation, e.g. in terms of thickness etc?

*As this is the results part, we do not discuss any implications for sample preparation. But we did follow the reviewer's suggestion by adding new text in Section 5.1 (Discussion) about sample preparation and inclusion picking procedure to avoid pressure release due to proximity to thin section surface.*

Line 223. "Assuming that the thin-section surface is sufficiently far away from a quartz inclusion and no microcracks appear around quartz inclusion..." I recommend replacing "microcracks" with "yield" as your solution does not consider microcracks and there is a discussion on microfracture later.

*Done.*

Line 227. "The flow law of garnet from Wang and Ji (1999) is applied". Please describe briefly this law.

*This is a good point that we have previously missed. Now a new equation has been added for Wang and Ji's flow law with numbers. (Eq. 21)*

Line 272. "The three mechanisms investigated here, i.e. viscous creep, plastic yield and proximity of inclusion..." Plastic yield was studied only together with creep, i.e. on a geological time scale. Plastic

yield without creep as might occur e.g. during sample preparation was not studied. Thus, I think it is more appropriate here to use term visco-plastic flow instead of plastic yield.

*Done. We thank the reviewer for pointing that out.*

Section 4.1. C is not cohesion, please check relevant values and your estimations for C.

*This point has been raised by the reviewer before and we agree. We have adjusted the terminology used here. We now refer C as the plastic yield strength. We have checked the entire manuscript to change cohesion into yield strength.*

Lines 283-289. "This suggests that plastic yield does not occur in an idealized scenario of isotropic, spherical quartz inclusion entrapped in infinite garnet host. However, such an ideal scenario is highly improbable in natural samples. The observed cracks in garnet host may be formed due to potential reasons including: 1) elevated differential stress when the inclusion is close to thin-section surface ("ring" shaped pattern in Fig.4a); 2) stress concentration at the corners of quartz inclusion (Whitney et al., 2000); 3) anisotropic elastic deformation of the quartz inclusion (e.g. Murri et al., 2018); 4) pre-fractures/weakness in garnet host before the entrapment of quartz inclusions."

While I agree with the possibility of elevated differential stress and stress concentrations at the corners, I would like to emphasize that this statement is based on the solution for materials obeying Tresca criterion, which does not describe fracturing. To make conclusions about fractures around inclusion, one needs to consider other failure criteria such as Griffith or Mohr-Coulomb, where cohesion and tensile strength play major role. Solutions for plasticity onset and failure pattern in elastoplastic and visco-plastic rocks with these failure criteria are available in the literature. They would give other estimations for pressure necessary to induce fractures.

*We agree with the reviewer on this point. Here, we have deleted the words related to cracks. The point is that we may still have localized stress due to the following reason stated above, hence localized dislocation around a non-spherical, anisotropic inclusion. Therefore, a statement on localized plastic yield around natural mineral inclusion is valid, just that we do not involve cracks in our discussion.*

Conclusions. "We presented a 1D visco-elasto-plastic model to study the inclusion-host system undergoing prograde/retrograde P-T path" There are at least two different models in this paper.

*We added "first" and "then" before the two sentences for the two models to separate them.*

"A simplified analytical solution for inclusion pressure (Eq. 32) close to stress-free thin-section surface is derived." The solution presented by authors was not new, it was re-derived after original paper by Seo and Mura [1979].

*We have rephrased this sentence so that it is clear that the solution was not original. The derivations were moved to the appendix.*

Please also make some statements on the implications for Raman-thermobarometry and how to use your results.

*This is a helpful comment. We have added some text in "Conclusions" about the petrological implications of our model and summarize what we have found about the determination of entrapment pressure of quartz inclusion and the application of garnet overstepping model.*

*In the end, we again thank the reviewer for all the detailed comments that greatly improved the quality and technical correctness of the manuscript.*

*General reply to reviewer 2*

*We thank the very positive feedback from the reviewer and helpful comments in the review letter. We have carefully considered all the comments given by the reviewer and made corresponding changes to the manuscript. In this reply letter, the original comments from the reviewer are in "*regular black*" and our replies are in "italic blue"*

I apologize for the delay of the review process but I had too many administrative issues to deal with first. The manuscript is actually one of those that make the review process very easy. It is well-written, it has well organized structure and the addresses a timely topic with a novel approach. On top what referee #1 has been already mentioned, I only have suggestions, basically no real criticism. I like the way the authors explain the methods in chapter 2. And finally discuss the most important findings of the study in a reasonable detail. I would only suggest to spent less words on the "distance to surface" issue (chapter 3.2) as this is not really new, but focusing more on the "over-" and "underestimation" of the pressure depending on the PT paths the samples take. This is really exciting and should be highlighted even more.

*We totally agree with the reviewer that the "distance to surface" issue has been studied in several papers cited in the manuscript. We took the advice and move the previous derivation part into the appendix. However, we would like to highlight that the new thing that we present is this very simple form of pressure distribution (now in Eq. 22). Compared to the cumbersome formulas of stress components in e.g. Seo and Mura (1979) and Mindlin and Cheng (1950), the pressure distribution can be simplified to a concise closed form. This is particularly relevant in studying the residual pressure distribution and pressure release due to proximity to thin-section surface. Also, when performing numerical model benchmark, this is particularly helpful because of its simplicity. If one wants to perform a fast validation on the numerical solution, our Eq. 22 can be easily used to compare with the pressure calculated with numerical code. Therefore, we leave the final equation for pressure distribution (Eq. 22) in the main text and most of the derivations in the appendix. The "distance to surface" part has been revised accordingly to clarify these points.*

To me the authors could consider to keep the inclusion-host relationship a bit broader in the

begin of the introduction, such as considering inclusions in a bit broader context (see Farber et al 2014 CMP, for instance), but this may result in a less sharp structure.

*We agree with the reviewer's point to make the text of wider appeal to general readers in petrology. Therefore, we have added several references including the suggested one at the beginning of the text to reach a broader audience.*